# Freight Wagon Digitalization for Condition Monitoring and Advanced Operation

**DOI:** 10.3390/s23177448

**Published:** 2023-08-27

**Authors:** Iker Moya, Alejandro Perez, Paul Zabalegui, Gorka de Miguel, Markos Losada, Jon Amengual, Iñigo Adin, Jaizki Mendizabal

**Affiliations:** 1CEIT-Basque Research and Technology Alliance (BRTA), Manuel Lardizabal 15, 20018 Donostia/San Sebastián, Spain; aperez@ceit.es (A.P.); pzabalegui@ceit.es (P.Z.); gdemiguel@ceit.es (G.d.M.); mlosada@ceit.es (M.L.); iadin@ceit.es (I.A.); jmendizabal@ceit.es (J.M.); 2Universidad de Navarra, Tecnun, Manuel Lardizabal 13, 20018 Donostia/San Sebastián, Spain

**Keywords:** railway, digitalization, freight, monitoring, wagon, infrastructure

## Abstract

Traditionally, freight wagon technology has lacked digitalization and advanced monitoring capabilities. This article presents recent advancements in freight wagon digitalization, covering the system’s definition, development, and field tests on a commercial line in Sweden. A number of components and systems were installed on board on the freight wagon, leading to the intelligent freight wagon. The digitalization includes the integration of sensors for different functions such as train composition, train integrity, asset monitoring and continuous wagon positioning. Communication capabilities enable data exchange between components, securely stored and transferred to a remote server for access and visualization. Three digitalized freight wagons operated on the Nässjo–Falköping line, equipped with strategically placed monitoring sensors to collect valuable data on wagon performance and railway infrastructure. The field tests showcase the system’s potential for detecting faults and anomalies, signifying a significant advancement in freight wagon technology, and contributing to an improvement in freight wagon digitalization and monitoring. The gathered insights demonstrate the system’s effectiveness, setting the stage for a comprehensive monitoring solution for railway infrastructures. These advancements promise real-time analysis, anomaly detection, and proactive maintenance, fostering improved efficiency and safety in the domain of freight transportation, while contributing to the enhancement of freight wagon digitalization and supervision.

## 1. Introduction

In the freight train sector, there is a significant lack of knowledge about the state of deterioration of the railway infrastructure and the trains themselves, mainly due to the absence of digitization and advanced monitoring. Reliant solely on manual inspections and visual assessments, the industry has faced significant challenges in optimizing maintenance and ensuring operational efficiency. However, recent advancements in freight wagon digitalization have brought about a paradigm shift in this sector.

The digitalization of freight trains is a crucial advancement aimed at creating modern functionalities that provide a cost-effective and appealing service, while also offering improved operational opportunities to operators and infrastructure managers. These modern functionalities encompass intelligence, detection, actuation, and communication capabilities.

Moreover, the digitalization of freight trains aligns closely with the principles of Industry 4.0, ushering in a new era of interconnected and intelligent systems. Embracing this transformative approach, the freight train sector can harness the power of well-established technologies such as sensor deployment and element virtualization. These technologies, already matured and successfully applied in domains like Industry 4.0, offer great potential for revolutionizing the railway industry [1]. By strategically integrating sensors, the digitalization process enables real-time monitoring of vital aspects such as train composition, train integrity, wagon asset condition, and continuous wagon positioning. The seamless communication facilitated by these advanced technologies fosters a data-driven ecosystem that empowers operators and infrastructure managers with valuable insights for enhanced decision-making and proactive maintenance strategies. Thus, the utilization of these mature technologies becomes a cornerstone in advancing the efficiency, reliability, and safety of freight services in the contemporary railway landscape.

According to its principles, the transport industry must significantly enhance the cost competitiveness and dependability of freight services to fulfil the ambitious goals outlined in the Transport White Paper [2] for the advancement of rail freight. These goals include nearly doubling rail freight usage compared to 2005, achieving a 30% shift of road freight over distances exceeding 300 km to modes such as rail or waterborne transport by 2030, and surpassing a 50% shift by 2050. Consequently, it is crucial to improve the cost-effectiveness and reliability of freight services to meet these objectives successfully.

Rail freight must adopt a cost-effective and appealing approach to entice shippers and divert freight from the congested road network. The challenge at hand entails two key aspects:Establishing a new service-oriented profile for rail freight services that prioritizes punctual deliveries at competitive prices. This entails integrating operations with other modes of transportation, incorporating innovative value-added services to cater to customer needs, and striving for operational excellence.Enhancing productivity by addressing existing operational and systemic weaknesses, including interoperability issues. This can be achieved by seeking cost-effective solutions, optimizing the utilization of current infrastructure, and embracing technology transfer from other sectors to enhance rail freight operations.

By addressing these challenges, rail freight can position itself as a reliable and efficient alternative, contributing to the shift of freight from the congested road network while providing a cost-effective and attractive service to shippers.

The freight railway environment presents a set of formidable challenges characterized by its extensive geographical distribution, harsh environmental conditions, and stringent energy considerations.

This article presents a comprehensive overview of the recent developments in freight wagon digitalization, focusing on the definition, development, and field tests conducted on a commercial line in Sweden. With the integration of a wide range of components and systems, the concept of the intelligent freight wagon has emerged. This digitalization process involves the strategic installation of sensors that enable various functionalities, including train composition, train integrity, wagon asset monitoring, and continuous wagon positioning. Furthermore, advanced communication capabilities facilitate seamless data exchange between these components.

To validate the effectiveness of this digitalization approach, field tests were carried out on three freight wagons operating on the operational line between Nässjo and Falköping in Sweden. These wagons were equipped to monitor the behavior of the train, enabling the detection of faults or anomalies in both the wagons and the railway infrastructure. This integrated approach not only enhances safety but also lays the foundation for a comprehensive monitoring solution for railway infrastructures, enabling real-time analysis, anomaly detection, and proactive maintenance.

The remaining sections of this article are organized as follows. First, the existing work on the digitalization of freight wagons is described (Section 2). Subsequently, a comprehensive overview of the developed system, including its services and functionalities, is provided (Section 3). Following that, the test campaign is presented, outlining the methods and procedures employed (Section 4). The results and discussions are then presented in Section 5, offering valuable insights and highlighting the significance of data acquisition. In Section 6, the main results and their implications are summarized, together with suggestions for future research and potential areas for improvement based on the findings.

## 2. Related Work

This section describes the related work for on-board monitoring for rolling stock and infrastructure condition determination found in the literature and in EU research projects.

Shift2Rail [3] is the first European rail initiative to seek focused research and innovation (R&I) and market-driven solutions by accelerating the integration of new and advanced technologies into innovative rail product solutions. Shift2Rail promotes the competitiveness of the European rail industry and meets changing EU transport needs. R&I carried out under this Horizon 2020 initiative develops the necessary technology to complete the Single European Railway Area (SERA).

One of the main objectives of TD3.8 Intelligent Asset Management Strategies (IAMS) is to shift towards a tailor-made maintenance approach by using the necessary tools for information management and decision support. This enhances the need to digitalize railway assets. Information is derived from the data obtained on board and on field. One of the most needed digitalizations is in the freight railway subsector, focused on the IP5 pillar for Shift2Rail. These activities are mostly based on the successful progress of TD5.1 fleet digitization and automation and mostly TD5.3 smart freight wagon concepts. For condition monitoring on the freight subsector, TD5.3.3 extended market wagons and TD5.3.4 telematics and electrification have made the greater efforts and they have been delivered on the documentation, demonstrations and results presented.

As for EU Rail, FP3 [4] and FP5 [5] are the pillars concerned and they have just started their activities, so there is no published information nor are there any conclusions related to railway onboard and infrastructure condition monitoring.

From the academic and scientific point of view, Figure 1 presents the time evolution of the research papers related to on-board monitoring for rolling stock and infrastructure condition determination. The increasing number of papers since 2016 proves the growing interest in this field in the past few years. It also shows that the technology and techniques are in the right place to serve the needs of the railway industry. Research works such as the one discussed in [6] show that the deployment of sensors on freight wagons allows, indeed, the detection and transmission of multiple status information regarding the maintenance and safety of these rolling elements.

The most cross-cited papers from the comprehensive list of references [7,8,9,10,11,12,13,14,15,16,17,18,19,20,21,22,23,24,25,26,27,28,29,30,31,32,33,34,35,36,37,38,39,40,41,42,43,44,45,46] represent the current state of the art in the field of condition monitoring for railway infrastructure. These articles primarily focus on advanced monitoring techniques and track quality assessment, including the findings of supervised experiments conducted on Polish railway lines using the electric multiple unit (EMU-ED74) equipped with a prototype track quality monitoring system [17]. The system incorporates a track quality indicator (TQI) algorithm, which utilizes a given transformation to preprocess the acceleration signals. This preprocessing is employed to extract the fundamental dynamics from the measured data, enabling a more comprehensive evaluation of the geometrical track quality. A comparative analysis is conducted to assess the performance of the proposed approach against other existing methods. This solution is the core of a track inspection system on board an electrified unit, which is a first step but it is not directly employable on freight wagons mainly due to power and location constraints. In addition, this paper presents further advanced features for freight train operation.

The second most cited [14], from 2021, is a survey which presents a comprehensive examination of the current literature conducted to provide an updated and content-driven analysis. This theoretical analysis is also of great interest to the research presented in this paper as it identifies the key contributors who have significantly influenced the progress of research in the specific area of interest. Using a coupled methodology that combines bibliometric performance analysis and a systematic literature review, the authors are able to identify the influential researchers, journals, and papers in the field. The findings of this study not only highlight the research trends pertaining to the analyzed area but also shed light on future research directions, particularly from an engineering standpoint. The main trends have also been considered in the research presented in this paper.

The following list discloses the different referrals and nuances collected by this research for the definition of “condition monitoring”:Direct measurement of relevant signals with time and/or frequency domain signal processing. Collection and real-time recording of digital and analogue signals using distributed transducers.Detecting and identifying deterioration in component structures and infrastructure performance in operating conditions. Continuous or periodic monitoring options.Alarm tool for maintenance. Distinguishing between normal and abnormal conditions and thresholding techniques for alarm systems.Implementing proactive condition monitoring technology. Tracking technical degradation and implementing preventive activities.Fault detection and diagnosis systems with intelligent algorithms. Condition-based monitoring for prognosis and diagnosis of component degradation.Ensuring safe and cost-effective train operation.Gathering and processing data for design, availability, reliability, and maintenance support.

Enhanced infrastructure monitoring of various elements such as bridges, viaducts, tunnels, crosses, rail gaps, frozen soil, and leaky feeders can yield significant benefits in terms of efficiency and safety. Neglecting safety and security monitoring of railway infrastructure poses risks such as train collisions, derailments, terrorism, and wagon failures. Notably, infrastructure or rolling stock failures still account for 35% of train delays, indicating the potential for substantial performance enhancements through intelligent systems in railway freight management [28].

From the alternatives listed above, refs. [40] and [23] categorize them into three levels here introduced and expanded in the picture below:Level 1 Data Logging and Event Recording Systems. When major incidents occur, they are used primarily to provide conclusive evidence. Equipment and operations are generally recorded digitally. This type of system can be used to detect faults in certain assets whose operation time or logic changes under fault conditions. Such systems are generally devoid of any data analysis. Typically, remote access is available to the systems, and data are logged locally.Level 2 Event Recording and Data Analysis Equipment. In addition to Level 1, this offers basic data analysis options, including statistical or sequence analysis. It is generally equipped with additional communication modules for remote access to data and analysis. In general, these systems are used for fault detection or the investigation of allegations but are unable to predict future failures.Level 3 Online Health Monitoring Systems. These systems are defined as the highest level of condition monitoring. These devices gathers digital and analogue (digitized) signals from monitored equipment, analyze them into characteristic signatures, compare them with an internal database of healthy and simulated faulty operation modes, and signal alarms and fault diagnosis information to operators. Expert systems, knowledge bases, and look-up tables are standard analysis techniques.

As a complement, Figure 2 illustrates an example of an intelligent infrastructure framework for railways [47]. It completes the level categorization with examples of uses and services that could be served with the equipment and strategy put in place for the monitoring.

The main conclusion from the analysis shown in this section is that a connected, distributed, and integrated system, with more layers and distributed acquisition and processing subsystems, is able to provide more useful information. The work presented in this paper is the result of the work performed in the TD5.3 smart freight wagon concepts topic and the result is part of the final demonstration performed as the conclusive activity for condition monitoring.

## 3. Perspective of the System: A High-Level Overview

The digitalization framework for freight wagons presented in this section is applicable to wagon assets and infrastructure monitoring and is designed to acquire and monitor data from various sensors installed on the wagons, enabling efficient and reliable operation. The monitoring with several sensors on each bogie is complemented with train composition, train integrity and positioning, which data are combined and converged for more accurate processing of the raw data. The visual representation, displayed in Figure 3, provides a clear and concise overview of the system’s architecture. It showcases the various components and their interconnections, offering a comprehensive understanding of how the system is designed. By referring to Figure 3, one can easily grasp the hierarchical structure, the flow of data, and the relationships between different modules within the system.

The locomotive on-board unit (LOBU) is responsible for controlling and enabling all communications. The LOBU serves as a central hub, storing and processing data received from all the connected wagons. It plays a crucial role in coordinating data exchange and ensuring seamless integration of information.

The system architecture comprises several hardware components that work together to enable data acquisition, storage, and analysis. Each freight wagon is equipped with a wagon on-board unit (WOBU), which serves as a local data storage and communication device. The WOBU collects and stores persistent information about the wagon, such as its identification, type, and available functionalities. It also acts as a gateway for sensor data acquisition.

The HW definition of the wagon on-board unit system deployed in each wagon is presented below; this is a connected multiprocessing platform. This architecture consists of two controllers for processing, which are a mainstream microcontroller (STM32F105RC) and a system on module (SOM) (iMX8-based), a series of devices for sensorization, communications and the power supply system [20].

A customized card was employed, featuring an SODIMM type connector, to interface with a VAR-SOM-MX8M-MINI [20]. It incorporates a certified railway connector, designed for railway applications, to enable the wiring of a CAN bus in addition to the Ethernet and USB interfaces. Figure 4 depicts the block diagram of the designed hardware (HW). The mechanical dimensions measure 150 mm × 95 mm × 45 mm.

Connectivity between the LOBU and WOBU, as well as between multiple WOBUs, is established through a scalable communication infrastructure. This ensures efficient data exchange and synchronization, enabling real-time monitoring and analysis capabilities.

In addition to the LOBU and WOBUs, the system includes a driver desk, which provides a user interface for direct connectivity to the on-board system. The driver desk allows for efficient interaction and communication with the system, facilitating control and monitoring of various functionalities.

Furthermore, the system incorporates the control center, a centralized platform for control and monitoring. The control center retrieves data from the cloud storage and enables remote monitoring and analysis of the acquired information. It serves as a comprehensive management tool, providing insights into train performance, wagon behavior, and infrastructure evaluation. Advanced algorithms can be applied within the control center to derive valuable conclusions and optimize decision-making processes.

The system encompasses essential functionalities such as train composition, train integrity, continuous wagon positioning, and spring monitoring. These functionalities play a crucial role in organizing wagons, ensuring connectivity and safety, tracking wagon location, and detecting spring faults.

In the following sub-sections, the specific functionalities implemented within the system are explored, providing detailed explanations of their capabilities and the benefits they offer for comprehensive freight wagon digitalization and condition monitoring.

### 3.1. Train Composition

This functionality is a pivotal aspect of freight train operations. It involves strategically organizing and assembling wagons to create an efficient transport unit. Its importance lies in optimizing various aspects of freight operations, such as weight distribution, load balancing, and overall train performance. By carefully arranging wagons and ensuring seamless connectivity between them, logistics managers can achieve optimal resource allocation, streamlined logistics processes, and enhanced operational efficiency. Additionally, efficient train composition reduces stress on rail infrastructure, minimizing wear and tear. Accurate identification and tracking of wagons within the train formation enable real-time monitoring, cargo identification, and efficient resource utilization.

In the developed system, each wagon is equipped with a wagon on-board unit (WOBU) that stores essential information such as wagon identification, type, number of bogies, and available functionalities. This WOBU functionality is triggered upon request from the driver desk, as mentioned earlier in this subsection. The driver desk initiates a discovery process through the LOBU (locomotive on-board unit), which communicates with the connected wagons. In response, each wagon provides persistent information along with the real-time status of the connected sensors. The LOBU processes these data to establish the current composition of the train and subsequently notifies the driver desk to display the updated train information as shown in Figure 5. This streamlined process ensures effective communication and seamless coordination between the different components of the system.

This advanced train composition functionality offers a comprehensive solution for managing the composition of freight trains. Leveraging digitalization technologies, our system provides real-time insights into train formation, enabling logistics operators to make informed decisions regarding load distribution, coupling order, and overall train configuration. This not only optimizes train performance but also enhances safety, reduces operational costs, and improves the overall efficiency of freight transportation.

### 3.2. Wagon Positioning

The integration of a position stamp in freight wagon monitoring services is crucial for accurate evaluation and efficient tracking. It provides timestamped records of wagon locations throughout their journey, enhancing safety, optimizing operations, and enabling digitalization in freight transportation.

Accurate and real-time location tracking is a primary reason for implementing a position stamp. It allows stakeholders to precisely track wagon locations, ensuring safety and enabling proactive measures in response to deviations or incidents. Real-time tracking also facilitates efficient resource allocation, optimized loading/unloading operations, and informed decision-making.

The use of a position stamp optimizes maintenance schedules and resource allocation. Continuous monitoring helps identify maintenance requirements, minimizing breakdown risks and maximizing operational efficiency. The data from position stamps provide insights into wagon utilization patterns, informing resource allocation decisions, routing optimization, and fleet management practices.

The integration of position stamps supports comprehensive digitalization. Time-stamped position data enable efficient documentation, data-driven decision-making, and advanced analytics. Leveraging this data, including machine learning algorithms, helps identify optimization opportunities, improve route planning, and enhance supply chain visibility.

To provide time-stamped position data, the proprietary hardware of WOBUs (wagon on-board units) employs single-frequency multi-constellation GNSS receivers. These receivers translate satellite signals into messages and estimated satellite receiver distances.

The algorithm utilizes GPS and Galileo observables to estimate WOBU positions along the train’s route. Due to suboptimal satellite visibility caused by the GNSS antennas’ lateral location between freight containers, a least squares estimation algorithm is employed. This algorithm allows recalculation of positions based on the required information, without considering past measurements or results. It provides positions despite harsh railway environments. Multiple WOBUs offer position redundancy for post-processing and error analysis. Figure 6 shows the driver desk visualization of the wagon positioning functionality.

### 3.3. Train Integrity

The monitoring of the integrity of a freight train has turned out to be an essential requirement to operate the train in a safe way. To have the confirmation of the integrity of the train, the operator ensures that the full train is travelling towards its destination and no goods have been left in the way. Moreover, if this system is used as part of the safety critical signaling system used for the operation of the railway, the occupancy of the lane can be increased due to knowledge of the position and completeness of the train and its wagons. This subsection introduces the different train integrity classes defined in the X2RAIL-4 project [48] and how they could be used alone, or in a combined way to ensure the integrity of the freight train.

X2RAIL-4 project defined three train integrity classes depending on the technology used to measure it:Train integrity class 1: This relies on wired net connectivity. All the wagons are wired, forming a net that goes from the locomotive to the tail of the train. The LOBU, placed in the locomotive, is continuously monitoring the wired composition functionality to verify that all the WOBUs connected at the beginning of the operation are still connected to the network. Any fault detected in the aforementioned network generates an alarm message in the train integrity class 1 function.Train integrity class 2: This relies on the coherence between the velocities measured at the head and tail of the train. The velocity of the train is continuously measured both at the head and tail of the train. These velocities are then compared. If there is a difference bigger than a threshold programmed for the lane in which the freight train is operating, a train integrity class 2 alarm is raised in the system.Train integrity class 3: This relies on the distance measured between wagons. The head and the tail of each of the wagons are equipped with an ultra-wideband (UWB) anchor. These anchors are used to calculate the distance between the tail of a wagon and the head of the next wagon. If the distance measured is less than the maximum distance between coupled wagons plus a security margin calculated depending on the maximum gap between wagons, a train integrity class 3 alarm is raised in the system.

The freight train can have deployed one or more of the introduced train integrity monitoring classes, as shown in Figure 7. In the case that only one of the classes has been installed, the whole train integrity function will be performed according to that class. If there is more than one class installed, the status of all of them will be taken into account, and the joint train integrity will be calculated taking into account the outputs of the existing classes and their probabilities of false alarms.

### 3.4. Wagon Monitoring System

Ensuring safe and efficient freight wagon operation relies heavily on monitoring springs. Springs play a vital role in absorbing shocks and vibrations, enabling a smooth ride while safeguarding cargo and wagon integrity. However, the springs endure extreme loads and adverse conditions throughout their service life. Factors like wear, fatigue, and severe impacts can lead to deterioration and loss of functionality over time, negatively impacting wagon performance and potentially leading to accidents.

Continuous monitoring of springs on freight wagons offers multiple benefits. First, it allows early detection of any deterioration or damage to the springs, which helps to prevent catastrophic failures and accidents. In addition, regular monitoring facilitates predictive maintenance, which means that springs can be replaced or repaired before serious problems occur. This not only improves safety, but also reduces operating costs by avoiding unplanned outages and optimizing maintenance schedules, minimizing disruptions, and ensuring a constant flow of goods. In addition, this spring monitoring functionality, working together with the wagon positioning function described above, facilitates the detection of possible faults in the track structure.

The spring monitoring functionality is performed by measuring the accelerations in the buffers of the bogies of the wagons, both at the top and at the bottom of the springs, as shown in Figure 8.

Measuring the accelerations that occur in this part of the wagon allows us to check whether the weight of the load in the wagon is balanced or not, to know the state of wear of the dampers themselves, as well as possible defects in the track infrastructure due to the vibration produced by the transit of the wheels of the wagon over possible imperfections in the infrastructure.

#### 3.4.1. Sensor Node HW Implementation

To collect data on the vibrations occurring in the springs, the development of the HW shown in Figure 9 was proposed.

This hardware consists mainly of an STM32F105RC microcontroller, which is responsible for managing the data measured by the two accelerometers proposed in this design. This controller incorporates an ARM Cortex-M3 32-bit RISC core operating at 72 MHz frequency. The two accelerometers proposed are the ADXL345 and the ADXL357, which allow one to select one or the other depending on the accuracy or sensitivity to be obtained in the measurements.

#### 3.4.2. Data Collection and Wagon Monitoring System Functionality Flow

To obtain the acceleration data at the springs, a network consisting of four sensor nodes connected through the CAN interface to the WOBU was deployed in every wagon. The arrangement of these sensor nodes in the wagon can be seen in Figure 10.

The data flow for the spring monitoring functionality is represented in Figure 11. Data from the three axes (x, y and z) is collected by the sensor nodes in 10-s time windows and sent to the WOBU through the CAN network. When the data is received at the WOBU, it stores it in its internal memory and sends it through the ETH network to the LOBU, which is in charge of storing all the information and processing the data coming from the wagons.

From the tablet, through a request to the LOBU, we can visualize the data as shown in Figure 12. The driver desk application on the tablet allows us to select which spring we want to monitor as well as to make a comparison between different springs and coordinate axes and different measurements within the same spring.

## 4. Test Campaign

This section provides a detailed description of the test campaign conducted to rigorously test and validate the functionalities outlined in the previous section. The test campaign was an integral part of the FR8RAIL-IV European project [49], aimed at evaluating the performance and effectiveness of the developed system. The campaign took place in Sweden from 22 May to 26 May 2023, and involved a planned route that encompassed various aspects of freight train operations. In this section, we will delve into the duration of the test campaign, the specific types of wagons utilized, the characteristics of the track, and the strategic placement of sensors, electronics, and antennas. These insights and findings from the test campaign are instrumental in assessing the reliability and efficiency of the digitalization and monitoring solution for freight wagons.

The journey, as shown in Figure 13, commenced at Nässjo station at 10 a.m., with the train arriving at Göteborg at 3 p.m. This allowed for the loading and unloading of containers in the wagons. At 6 p.m., the train departed from Göteborg, reaching Falköping at 10 p.m.

The average speed throughout the journey was maintained at 35 km/h, and the train made several stops at intermediate stations. The route encompassed various track sections, including track changes and a mid-journey train orientation reversal.

Figure 14 presents the freight wagons employed in the test campaign “Sggrss 80’|6-axle articulated intermodal wagon”. The train consisted of 21 wagons, with wagons 15, 16, and 17 being selected for monitoring, each carrying two containers. This strategic selection enabled comprehensive data collection for analysis.

In terms of hardware placement, as depicted in Figure 15, the electronic components and antennas were carefully installed in the middle section of the wagons, specifically in the stairwell area. This location ensured easy accessibility for maintenance purposes for the validation phase of the functionalities.

The test campaign provided valuable insights into the performance and functionality of the developed system under realistic operational conditions. The collected data serve as a crucial foundation for further analysis, validation, and enhancement of the digitalization and monitoring solution for freight wagons.

## 5. Results and Discussion

The primary objective, as mentioned earlier, was to acquire a substantial amount of data from various functionalities and establish an effective monitoring system, conditions included, with the intention of conducting comprehensive analysis in the future. The following are the results obtained for each functionality:Train composition: The system successfully obtained and displayed real-time information about the train, its wagons, and the connected sensors. The data acquisition and visualization were performed accurately, enabling efficient monitoring of the train’s composition, which is a key feature for train operation but also for the processing of the data incoming from the sensors.Train integrity: Continuous checks were carried out to ensure the train’s integrity while in motion. While the system effectively detected integrity breaches, there were occasional false positives at very low speeds. This aspect is being addressed for further improvement.Train positioning: Real-time visualization of the train’s current location and its individual wagons was achieved. The system provided accurate positioning information, allowing for effective monitoring and tracking of the train’s movement and its derivative on components and infrastructure monitoring.Wagon monitoring system: The continuous monitoring and real-time visualization of the accelerometers (both upper and lower) for each spring shown in this paper, provide valuable insights into the dynamic behavior of the wagon throughout its operation. By analyzing the combined data from these accelerometers, it becomes possible to assess the wagon’s response to the condition of the railway infrastructure. Figure 16 and Figure 17 illustrate the recorded data, offering a comprehensive understanding of how the wagon interacts with the track, thereby facilitating effective maintenance planning and optimizing the overall performance of the system.

Every operational data acquisition was logged with its corresponding timestamp and associated position. This detailed recording ensures that any faults, defects, or alarms can be precisely located and identified, facilitating prompt action and maintenance interventions.

During the validation tests, the train was constantly monitored through various means, including direct and remote connections from the driver desk, as well as from a centralized control center. With this feature deployed, the comprehensive monitoring approach ensures continuous oversight of the train’s operations, allowing for a quick response to any anomalies or emergencies.

All the collected information is also stored in a remote server, ready for future processing. This long-term storage enables thorough analysis and processing of the data to derive meaningful insights, contributing to enhanced operational efficiency and informed decision-making.

The obtained results not only validate the successful achievement of the objectives but also lay a robust foundation for conducting future in-depth analysis and deriving valuable insights from the accumulated data. Depending on the specific focus of the future analysis, this rich dataset can be instrumental in detecting anomalies, failures, and wear and tear, both within the wagons and across the track infrastructure. By leveraging this comprehensive monitoring approach, we can see significant potential for enhancing maintenance strategies, identifying potential issues proactively, and optimizing the overall performance and safety of both the rolling stock and the track system.

## 6. Conclusions

Based on the comprehensive analysis and findings presented in this study, the following key conclusions can be drawn:

First and foremost, the successful development and implementation of the digitalization system for freight wagons have not only addressed the limitations of traditional operational and manual inspections but have also showcased the immense potential for enhancing the monitoring and management of components and railway infrastructures. By integrating advanced sensors and monitoring technologies, the system has enabled accurate and real-time monitoring of train composition, train integrity, wagon asset monitoring, and continuous wagon positioning.

The primary objective of the system is comprehensive data gathering and monitoring. Collaborating with stakeholders, research institutions, and the railway industry is crucial for successful digitalization implementation. This approach brings diverse perspectives, enhances understanding of industry needs, optimizes resource utilization, and accelerates innovation deployment. By tailoring the system to specific demands, safety standards, and regulations, its overall effectiveness and acceptance in the industry are enhanced. Collaborative efforts foster knowledge exchange, driving further advancements in digitalization strategies to meet evolving freight transportation needs, ensuring informed decisions, improved efficiency, and enhanced safety.

Furthermore, the validation and testing campaign conducted on the operating line in Sweden provided crucial information on the real-world performance and functionality of the system. The data collected during the campaign served as the basis for the subsequent analysis, validation and improvement of the freight car digitization and monitoring solution. This data-driven approach allows for continuous improvement and optimization of the system, resulting in increased efficiency and reliability. The success of freight car monitoring in harsh and inaccessible environments demonstrates the system’s adaptability and its potential to provide comprehensive monitoring capabilities in a variety of operating conditions.

In addition, it is important to recognize the growing significance of effective data management in the context of large-scale digitalization efforts. As the volume of data generated by the digitalization system increases, adopting advanced data management techniques, such as big data analytics and machine learning algorithms, becomes essential. These cutting-edge technologies offer invaluable insights and opportunities for predictive maintenance, optimized resource allocation, and enhanced overall system performance. Therefore, the future direction of this research should focus on exploring these areas further to leverage the full potential of digitalization in freight wagon monitoring. By integrating big data analytics and machine learning algorithms, the system’s capabilities can be greatly enhanced, enabling proactive maintenance practices, and ultimately leading to improved operational efficiency and cost-effectiveness.

The monitoring of multiple wagons enables a thorough assessment of individual wagon behavior. Moreover, the combination of data from these wagons offers the opportunity to extract valuable insights regarding the overall condition of the railway infrastructure. Analyzing patterns and trends derived from the collective data can help identify potential defects or issues in the track infrastructure, enhancing maintenance planning and ensuring optimal system performance.

In conclusion, the digitalization of freight wagons and the integration of advanced monitoring capabilities offer transformative potential for the railway industry. By harnessing the power of real-time data, stakeholders can optimize operational efficiency, enhance safety measures, and improve the overall performance of railway infrastructures. As technology continues to advance, the successful implementation of digitalization in the freight wagon industry requires addressing emerging challenges. Ensuring interoperability among different systems, prioritizing data security and privacy, investing in research and innovation, and providing comprehensive training for stakeholders are essential steps. By proactively tackling these challenges, the industry can unlock the full potential of digitalization, leading to improved safety, efficiency, and overall performance of the railway infrastructure.

## Figures and Tables

**Figure 1 sensors-23-07448-f001:**
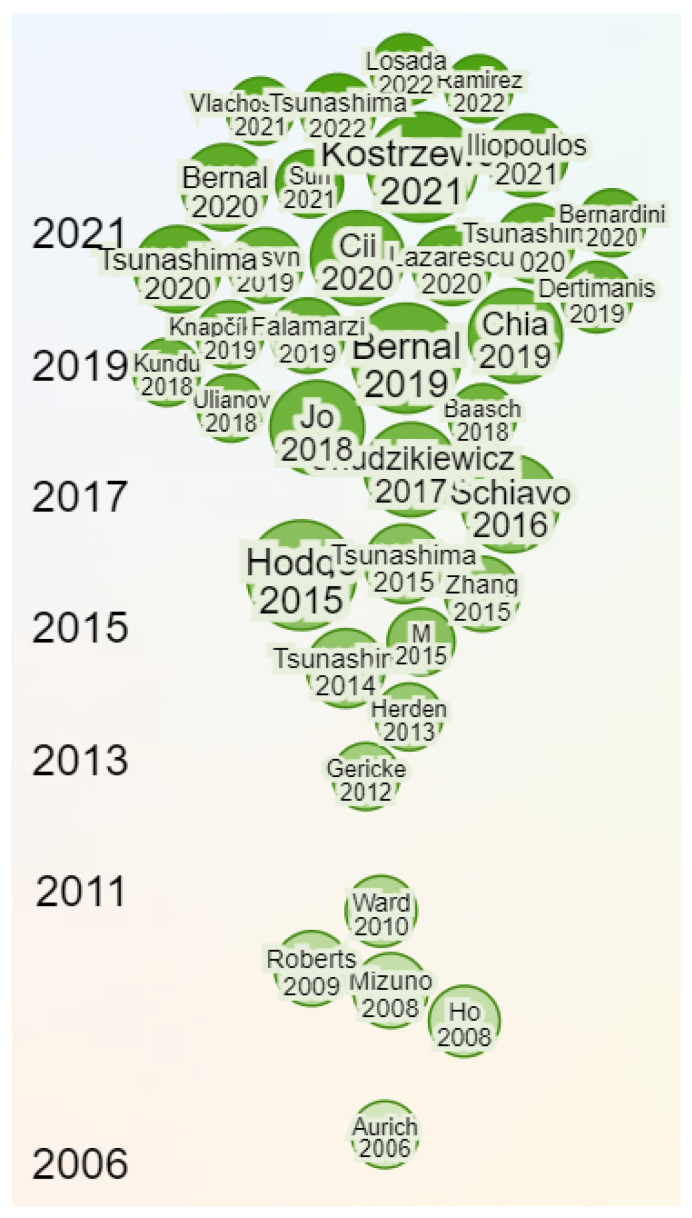
Time evolution of the research papers related to on-board monitoring of rolling stock and infrastructure condition determination [7,8,9,10,11,12,13,14,15,16,17,18,19,20,21,22,23,24,25,26,27,28,29,30,31,32,33,34,35,36,37,38,39,40,41,42,43,44,45,46].

**Figure 2 sensors-23-07448-f002:**
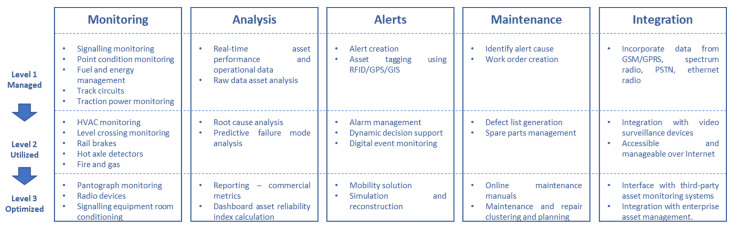
Intelligent infrastructure framework for railways [47].

**Figure 3 sensors-23-07448-f003:**
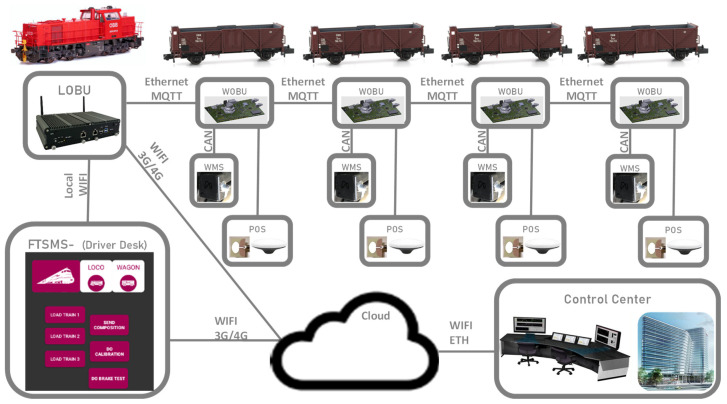
Architecture of the digitalization framework for freight wagons.

**Figure 4 sensors-23-07448-f004:**
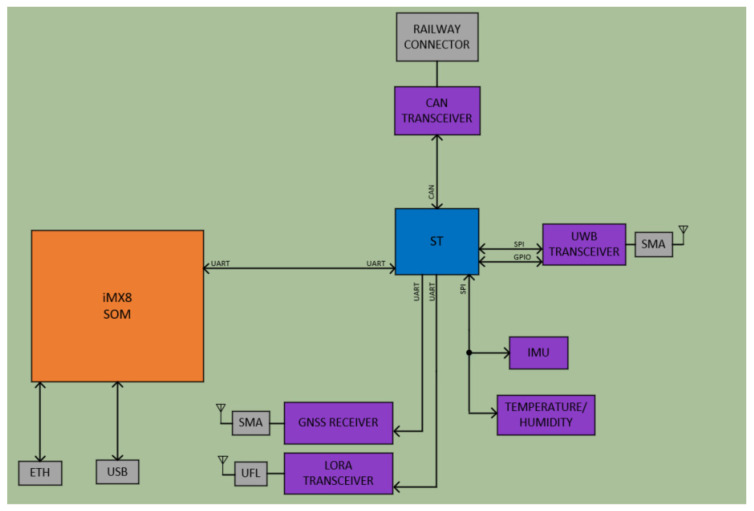
Block diagram of the WOBU proposed HW [10].

**Figure 5 sensors-23-07448-f005:**
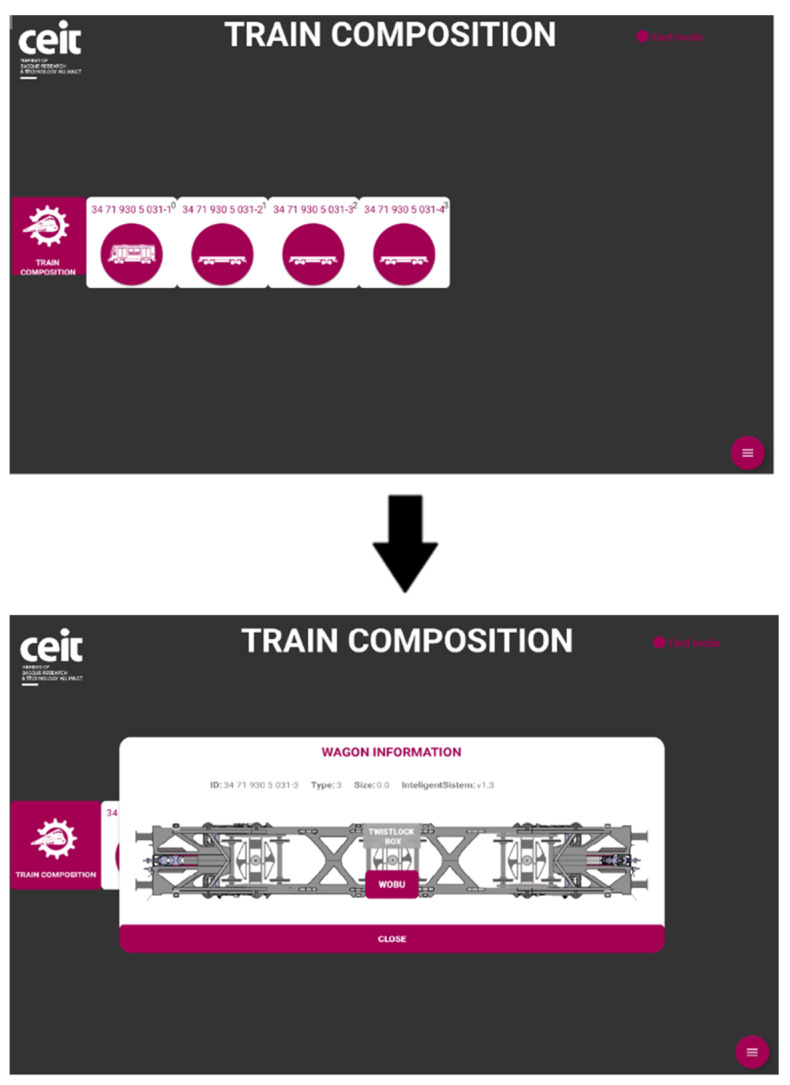
Driver desk visualization of the train composition.

**Figure 6 sensors-23-07448-f006:**
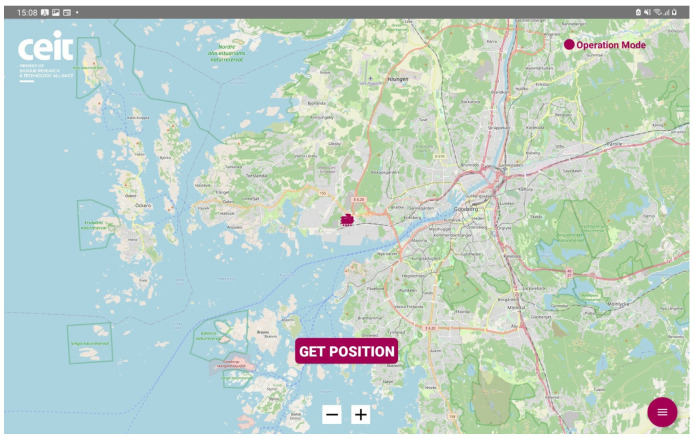
Driver desk visualization of the wagon positioning.

**Figure 7 sensors-23-07448-f007:**
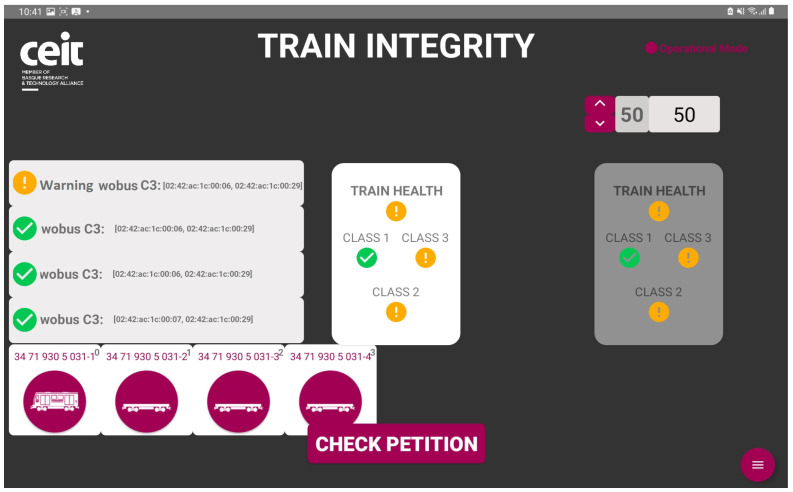
Driver desk visualization of the train integrity.

**Figure 8 sensors-23-07448-f008:**
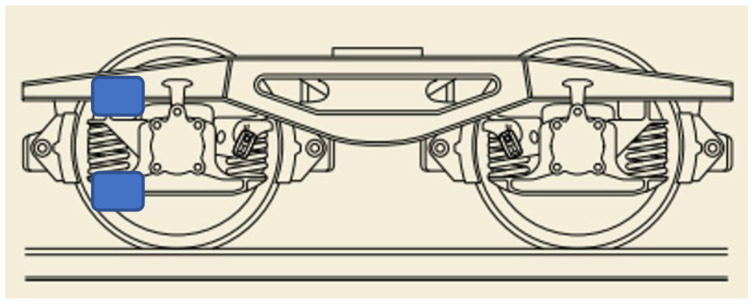
Vibration monitoring points on springs.

**Figure 9 sensors-23-07448-f009:**
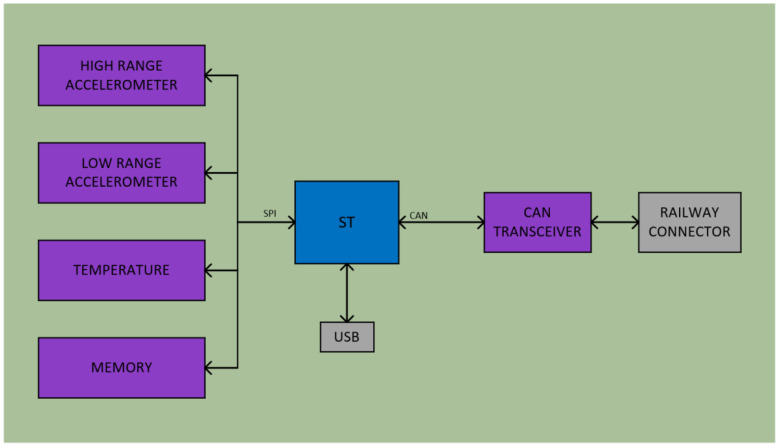
Block diagram of the sensor node proposed HW.

**Figure 10 sensors-23-07448-f010:**
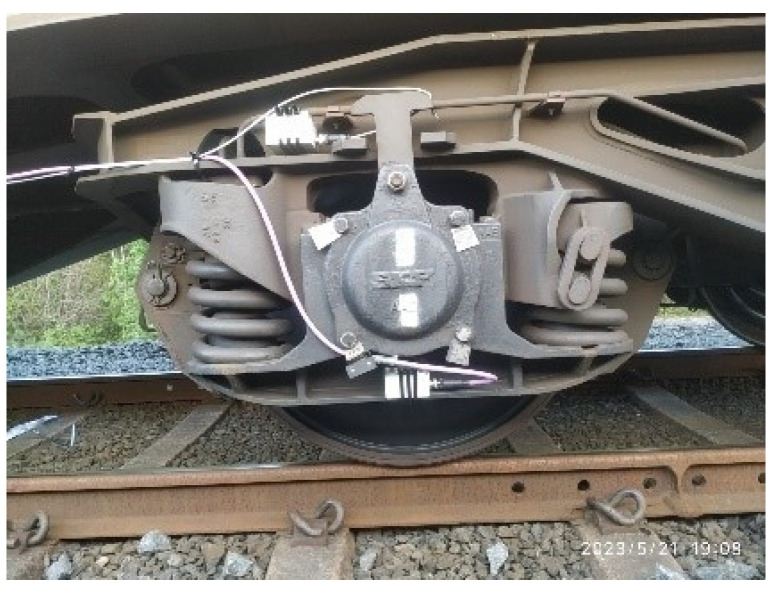
The arrangement of sensor nodes in the wagon.

**Figure 11 sensors-23-07448-f011:**
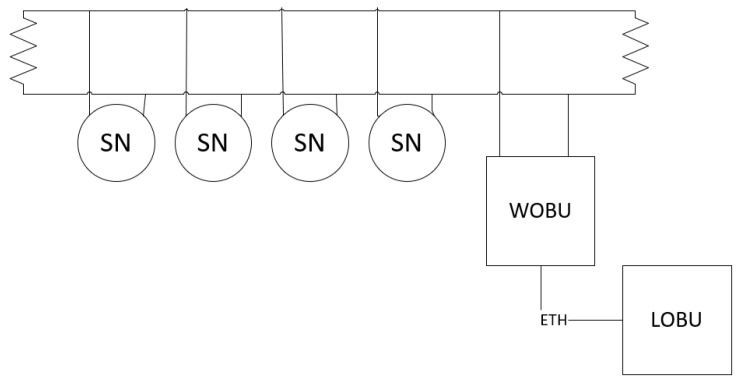
Schematic of the hardware involved in collecting acceleration data at the springs.

**Figure 12 sensors-23-07448-f012:**
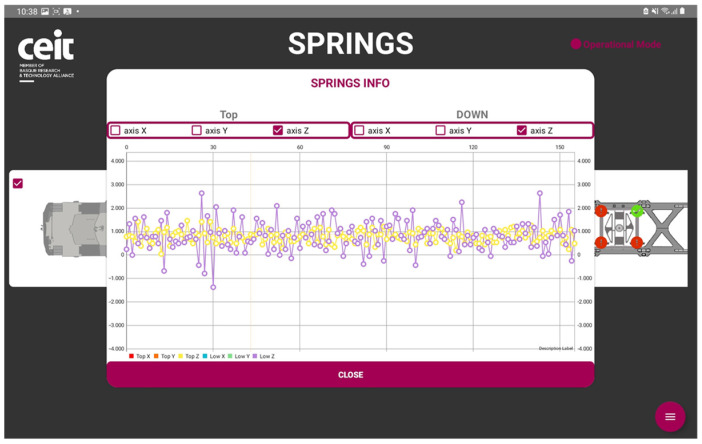
Driver desk visualization of the wagon monitoring system measurements.

**Figure 13 sensors-23-07448-f013:**
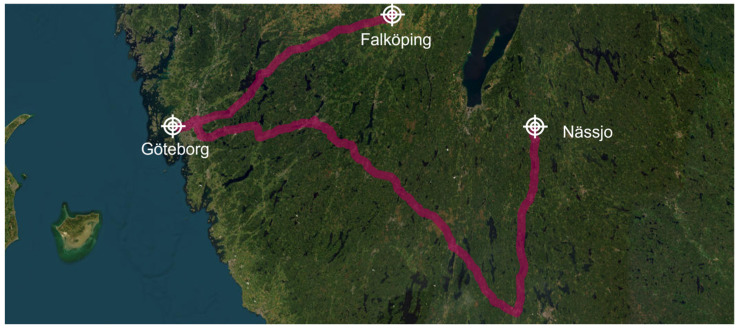
Journey undertaken by the freight train during the test campaign between Nässjo and Falköping.

**Figure 14 sensors-23-07448-f014:**
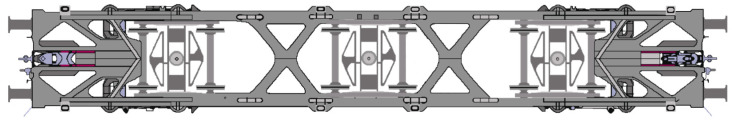
Schematic of the Sggrss 80´ | 6-axle articulated intermodal wagon [50].

**Figure 15 sensors-23-07448-f015:**
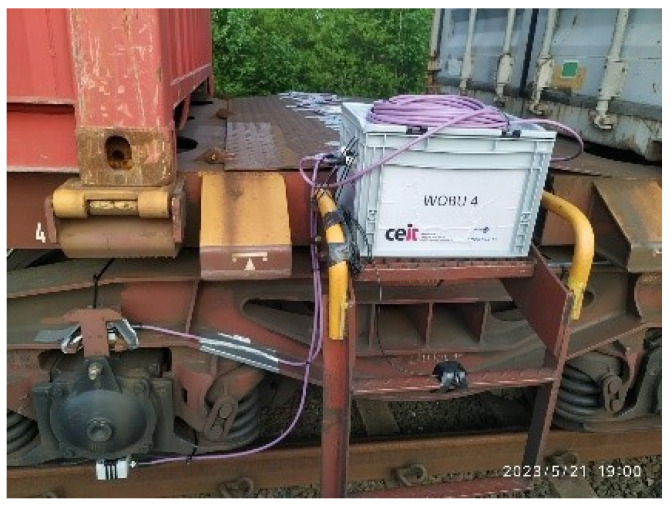
Installation of electronic components and antennas.

**Figure 16 sensors-23-07448-f016:**
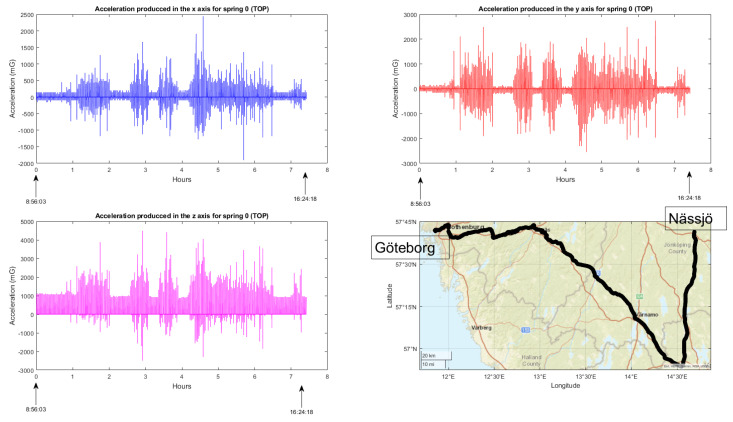
Accelerometer measurements above the wagon springs, displaying acceleration values in the x, y, and z axes together with travel route information.

**Figure 17 sensors-23-07448-f017:**
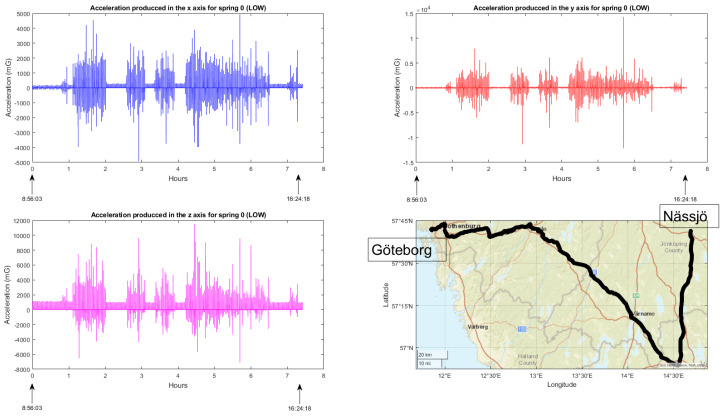
Accelerometer measurements under the wagon springs, displaying acceleration values in the x, y, and z axes together with travel route information.

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
