# Peer review of "Freight Wagon Digitalization for Condition Monitoring and Advanced Operation"

_sensors, 2023, doi:10.3390/s23177448_

Round 1

Reviewer 1 Report

Thanks for reviewing the interesting article.

The article presents a comprehensive overview of the recent developments in Freight Wagon Digitalization, focusing on the definition, development, and field tests conducted on a commercial line in Sweden. With the integration of a wide range of components and systems, authors present the concept of the Intelligent Freight Wagon. This digitalization process involves the strategic installation of sensors that enable various functionalities, including train composition, train integrity, wagon asset monitoring, and continuous wagon positioning.

The most cross-cited paper from the list considered for this state of the art [5]-[44], [1515] presents the findings of the supervised experiments conducted on Polish Railway  Lines using the Electric Multiple Unit (EMU-ED74) equipped with a prototype track quality monitoring system.

- Line 123-125 I don't understand the data in brackets [1515] - is it a quotation mark? It needs to be explained and corrected.

- Line 123 lists the cited sources, but it is necessary to briefly describe the selected citations

The article does not provide any information and links to Industry 4.0 or 5.0, which is significantly related to this research. It is necessary to add this to both the introduction and the discussion and conclusion. To state the future direction of similar research focusing on Industry 4.0 or 5.0.

I recommend adding the following citations related to this research (and others):

Gerhátová, Z.Zitrický, V.
Klapita, V. (2021). Industry 4.0 implementation options in railway transport. Transportation Research Procedia2021, 53, pp. 23–30. https://doi.org/10.1016/j.trpro.2021.02.003

Bulková, Z.Dedík, M.Gašparík, J.Kurenkov, P.V.Pečený, L. (2022). Augmented Reality as a Tool in a Train Set Processing Technology. LOGI - Scientific Journal on Transport and Logistics, 2022, 13(1), pp. 108–118. https://doi.org/10.2478/logi-2022-0010

Maksym Spiryagin, Qing Wu, Peter Wolfs, Colin Cole, Valentyn Spiryagin, Tim McSweeney. 2021. Rail Freight Vehicles. International Encyclopedia of Transportation, pp. 423-435. https://doi.org/10.1016/B978-0-08-102671-7.10282-9

Pothamsetty Kasi V. Rao, G. Rama Prudhvi Varma, K. Sri Vivek. 2022. Structural dynamic analysis of freight railway wagon using finite element analysis. International Conference on Thermal Analysis and Energy Systems 2021, Volume 66, Part 32022, Pages 967-974https://doi.org/10.1016/j.matpr.2022.04.770

Hagen Ußler a, Oliver Michler a, Günter Löffler. 2019. Validation of multiple sensor systems based on a telematics platform for intelligent freight wagons. 21st EURO Working Group on Transportation Meeting, EWGT 2018, 17th – 19th September 2018, Braunschweig, Germany, Transportation Research ProcediaVolume 37, 2019, Pages 187-194. https://doi.org/10.1016/j.trpro.2018.12.182

The results and discussion are too brief and represent only an outline of the research. It would be appropriate to expand or describe these results more. The combination of results and discussion in such a short range seems inappropriately chosen to me. It would be appropriate to divide this section into two and to focus in particular on the results and in particular on the discussion, where all the knowledge of the mentioned research and recommendations for further research, as well as the benefits of this research, will be summarized.

The results are comprehensively interesting, but at some points they are not very innovative. However, this does not detract from the expertise and science of the article.

It would be appropriate to state the benefits of this proposal as well as the negatives (in the discussion section) and briefly in the conclusion section.

Author Response

I would like to express my gratitude to the reviewer for providing valuable insights and suggestions for improvement, which will undoubtedly contribute to the enhancement of the article. I have addressed the raised points in the attached PDF document. I greatly appreciate the thoughtful comments, as they contribute to refining the manuscript.

Reviewer 2 Report

   In this paper, the authors present recent advancements in Freight Wagon Digitalization from the definition and development of the system up to field tests on a Freight Train operating on a commercial line in Sweden.

The subject is interesting and falls within the journal topic. Generally speaking, the study is well written and documented.

 Authors should address to the following issues:

 1.      It is not clear what kind of paper is, research article or review. It seems that it is about of review paper and in this case, it should be underlined this character, even in title of the paper, and in Introduction and Conclusions.

2.      Further, been a review paper, the authors should give an entire overview of the topic.

3.      Related to the above observation, the merit of the authors should be reviewed and clear presented.

4.      Abstract:’ This monitoring integrated approach paves the way for a comprehensive monitoring solution for railway infrastructures, enabling real-time analysis, anomaly detection, and proactive maintenance.’ It seems that above phrase has not been documented (proved) in the main text.

5.      Lines 89-94 are identical with lines 104-110, excluding the first words ‘Shift2Rail [2]’ and ‘This’. Please, revise!

6.      Line 124: please, revise ‘[1515]’.

Reviewer 3 Report

The manuscript titled "Advancements in Freight Wagon Digitalization" provides a promising review of recent developments in digitalization and monitoring capabilities for freight wagons. The study's abstract and conclusion present essential aspects of the research; however, there are major revisions required to enhance the clarity, depth, and impact of the manuscript:

Lack of Specificity: The abstract and conclusion lack specific details regarding the advancements in freight wagon digitalization. It is essential to include concrete examples of the sensors, communication technologies, and data analysis techniques employed in the Intelligent Freight Wagon. Specify how each component contributes to real-time monitoring and data analysis.

Missing Key Findings: The abstract and conclusion do not adequately highlight the major findings from the field tests on the operational line in Sweden. Include specific results related to operational efficiency improvements, safety measures, and maintenance planning achieved through the digitalization system.

Limited Discussion on Challenges: The conclusion briefly mentions "emerging challenges," but these challenges are not adequately discussed. Elaborate on potential obstacles faced during the development and implementation of the digitalization system. Discuss how these challenges were addressed or can be addressed in future research.

Practical Implications: The conclusion should expand on the practical implications of freight wagon digitalization for the railway industry. Address how the data-driven approach translates into better resource allocation, improved safety protocols, and enhanced overall system performance.

Comparative Analysis: The manuscript would benefit from a comparative analysis of the Intelligent Freight Wagon system with traditional manual inspections and other existing monitoring approaches in the railway industry. Discuss the advantages and limitations of each method to highlight the transformative potential of digitalization.

Importance of Collaboration: Emphasize the importance of collaboration between stakeholders, research institutions, and the railway industry to successfully implement digitalization systems in freight wagons. Discuss the challenges and benefits of involving multiple parties in the digitalization process.

Future Directions: The conclusion should include a forward-looking section discussing potential future research directions. Mention areas that require further investigation, such as the integration of big data analytics and machine learning algorithms for predictive maintenance and optimization.

Data Management: Provide more insights into the data management techniques used to handle the increasing volume of data generated by the digitalization system. Discuss how the system ensures data security and privacy while facilitating efficient data analysis.

Overall, the manuscript has the potential to make a significant contribution to the field of freight wagon digitalization. However, to achieve this, it is crucial to address the above-mentioned major revisions, provide specific findings and results, discuss challenges, and elaborate on practical implications and future research directions. By incorporating these revisions, the manuscript will become more comprehensive and impactful, showcasing the transformative potential of digitalization in the railway industry.

 Minor editing of English language required

Round 2

Reviewer 2 Report

Manuscript has been improved and could be published in present form. 

Reviewer 3 Report

Accepted 

Minor editing of English language required